# Genome-Wide Identification and Characterization of Short-Chain Dehydrogenase/Reductase (SDR) Gene Family in *Medicago truncatula*

**DOI:** 10.3390/ijms22179498

**Published:** 2021-08-31

**Authors:** Shuhan Yu, Qiguo Sun, Jiaxuan Wu, Pengcheng Zhao, Yanmei Sun, Zhenfei Guo

**Affiliations:** College of Grassland Science, Nanjing Agricultural University, Nanjing 210095, China; 2018220001@njau.edu.cn (S.Y.); 2017220007@njau.edu.cn (Q.S.); 2019120002@njau.edu.cn (J.W.); 2016220006@njau.edu.cn (P.Z.); 2019220006@njau.edu.cn (Y.S.)

**Keywords:** abiotic stress, *cis*-acting elements, expression profiles, *Medicago truncatula*, short-chain dehydrogenase/reductase

## Abstract

Short-chain dehydrogenase/reductase (SDR) belongs to the NAD(P)(H)-dependent oxidoreductase superfamily. Limited investigations reveal that SDRs participate in diverse metabolisms. A genome-wide identification of the SDR gene family in *M. truncatula* was conducted. A total of 213 *MtSDR* genes were identified, and they were distributed on all chromosomes unevenly. MtSDR proteins were categorized into seven subgroups based on phylogenetic analysis and three types including ‘classic’, ‘extended’, and ‘atypical’, depending on the cofactor-binding site and active site. Analysis of the data from *M. truncatula* Gene Expression Atlas (MtGEA) showed that above half of *MtSDRs* were expressed in at least one organ, and lots of *MtSDRs* had a preference in a tissue-specific expression. The *cis*-acting element responsive to plant hormones (salicylic acid, ABA, auxin, MeJA, and gibberellin) and stresses were found in the promoter of some *MtSDRs*. Many genes of *MtSDR*7*C,*
*MtSDR*65*C**, MtSDR*110*C**, MtSDR*114*C*, and *MtSDR108E* families were responsive to drought, salt, and cold. The study provides useful information for further investigation on biological functions of MtSDRs, especially in abiotic stress adaptation, in the future.

## 1. Introduction

Short-chain dehydrogenases/reductases (SDRs) belong to the NAD(P)(H)-dependent oxidoreductase protein superfamily [1,2]. This old family of metabolic enzymes is present in all organisms [3,4]. Although SDR proteins show diversity in structure and function, they have three common properties: (1) a cofactor binding site (TGxxxGxG), (2) a catalytical residue motif YxxxK, and (3) a conserved three-dimensional structure made up of “Rossmann fold” a-sheet with a-helices on both sides [5]. The SDR superfamily is usually divided into five types: ‘classical’, ‘extended’, ‘atypical’, ‘unknown’, and ‘divergent’, respectively, based on the structural characteristics [1,6].

A limited investigation on SDR in plants revealed that the SDR superfamily is involved in either primary or secondary metabolisms, such as fatty acid synthesis and elongation [7,8], chlorophyll biosynthesis or retrogradation [9,10], and terpenoids [11], steroids [12], phenolics [13], and alkaloids synthesis and metabolism [14]. The secondary metabolites of plants can be accumulated in response to stress in the form of chemical defense to provide a protective barrier. *AtSDR1*, being named as ABSCISIC ACID DEFICIENT2 (*ABA2*), catalyzes the multistep reaction from xanthoxin to abscisic aldehyde that is the key step of ABA biosynthesis [15,16,17]. *AtSDR3* is homologous to *AtSDR1,* which confers resistance to *Pst* DC3000 by regulating *AtPR-1* gene expression in a salicylic-acid-dependent pathway [18]. *CaMNR1* (menthone: (1)-(3S)-neomenthol reductase) catalyzes menthone reduction to generate neomenthol using NADP as a cofactor. Overexpressing *CaMNR1* improves the immunity to both bacterial and fungal pathogens in *Capsicum annuum* [19]. However, the role of SDR family members in abiotic stress adaptation has not been reported.

Legumes are important for both humans as food and livestock as feed [20,21]. *Medicago truncatula* is a model plant of legumes due to its modest genome size, diploid genome, and annual and autogamous nature, and it is easy to be transformed [22,23,24]. Given the potential importance of the SDR family, the genome-wide identification of *SDR* genes was conducted in *M. truncatula* in the present study to understand their gene structure, phylogenetic relationship, and common protein features as well as tissue-specific expression and responses to abiotic stress. The investigation will provide basic information for further comprehensive approaches on diverse functions of SDR in *M. truncatula* in the future.

## 2. Results

### 2.1. Identification and Classification of MtSDR Genes

With 178 Arabidopsis SDR protein sequences as baits, BLAST (Basic Local Alignment Search Tool) was applied to search for the target sequence of *M. truncatula* in order to comprehensively identify the SDR coding genes of *M. truncatula*. We used the hidden Markov model (HMM) profile of SDR (Pfam: PF00106, PF01370, and PF01073) [25] from the Pfam database to search the protein sequences in *M. truncatula*. The candidate SDR-encoding genes were manually chosen and functionally annotated based on the closest *A. thaliana* homology. Finally, a total of 213 *MtSDRs* were identified and were categorized into 42 families.

The gene locus ID, protein sequence length (SL), molecular weight (MW), and isoelectric point (pI) were summarized in Appendix A. Most of *MtSDRs* (88%) were divided into three main types, ‘classical’, ‘extended’, and ‘divergent’, while a minority (10%) was classified into unknown or atypical types. Five *MtSDRs* (*Medtr3g101500.1*, *Medtr4g086400.1*, *Medtr4g086410.1*, *Medtr6g088500.1,* and *Medtr8g044240.1*) showing up as homologous to *At4G333601.1*, *At1G49671.1,* and *At4G132501.1* had low HMM scores and no structural data. MtSDR proteins had the average length of 340 amino acids with arrangement from 185 amino acids for *MtSDR114C6* to 799 amino acids for *MtSDR2E1*. The MWs varied from 20.41 kDa (*MtSDR114C6*) to 89.92 kDa (*MtSDR2E1*), with an average of 37.40 kDa. The pI of the MtSDRs varied from 4.69 (*MtSDR114C11*) to 9.96 (*MtSDR50E3*), with an average of 7.14 (Appendix A).

### 2.2. Phylogenetic Relationships of MtSDR Proteins

To explore the evolutionary relationship among MtSDR proteins, the unrooted phylogenetic tree of 213 MtSDRs was aligned using MEGA-X software based on the sequences of MtSDR proteins (Appendix A). The chromosomal location and gene function annotation of *MtSDRs* was analyzed using TBtools software. Based on the phylogenetic tree, MtSDRs were classified into three primary groups (classical SDR fold, atypical members, and extended SDR fold) and seven subgroups, with 115 MtSDRs in classical SDR fold, 17 MtSDRs in atypical, and 81 MtSDRs in extended SDR fold (Figure 1, Appendix A). Moreover, the MtSDRs in the classical SDR fold were further classified into three subgroups (cluster C1, 2, and 3) consisting of 50, 45, and 20 members, respectively. The MtSDRs in extended SDR fold were further classified into three subgroups (cluster E1, 2, and 3) consisting of 18, 28, and 35 members, respectively (Figure 1 and Appendix A).

### 2.3. Ten Conserved Motifs Were Examined in MtSDR Proteins

We searched for the conserved motifs in 213 MtSDR proteins aiming to evaluate the functional regions of MtSDR proteins. Ten conserved motifs, namely motifs 1 to 10, were identified and indicated in the colored boxes based on their scale with sizes varying from 14 to 39 aa residues in width (Figure 2). Motif 1 encodes the conserved cofactor-binding site of ‘classic’, ‘extended’, and ‘divergent’ domains, motif 2 encodes the conserved active site of ‘classical’, ‘extended’, and ‘atypical’ domains. Motif 1 and motif 2 were observed in all *MtSDRs* (Appendix A). The other conserved motifs (motifs 3 to 10) also existed in MtSDR proteins with varying numbers from one to eight.

### 2.4. Chromosomal Location (Distribution) and Expansion (Duplication) Analysis of MtSDRs

In order to understand the distribution of *MtSDRs* in the genome, BLAST was used to locate 213 *MtSDR* genes on the corresponding chromosomes. *MtSDRs* were present on all chromosomes, while the arrangement and density of *MtSDRs* on every chromosome were uneven (Figure 3). The maximal number of *MtSDRs* (21.6%) was located on chromosome 4, whereas the minimum number of *MtSDR*s (6.1%) was located on chromosome 6. There was a high density of *MtSDR*s located on chromosomes 3, 4, 7, and 8.

The duplication patterns (tandem and segmental repeats) were analyzed using BLASTp and MCScanX (Multiple Collinear Scan toolkit). Twenty gene pairs were found to be paralogous in *MtSDRs*, and they belonged to segmental duplications (Appendix A). In addition, no tandem duplication was observed. The results showed that segmental duplication was the crucial driving strength for the evolution of *MtSDRs*.

Collinearity diagrams among *MtSDR*s were analyzed using gene duplication analysis. A total of 20 pairs of gene duplications were identified (Figure 4). The orthologous relationship between 213 *MtSDR* genes and 178 *AtSDR* genes from *Arabidopsis* or 315 *GmSDR* genes from *Glycin max* was analyzed. The results showed that there were 83 pairs of orthologous *SDRs* between *M. truncatula* and *A. thaliana* and 275 pairs between *M. truncatula* and *G. max* (Figure 4). The synonymous substitution rate (Ks) and non-synonymous substitution rate (Ka) of the *MtSDR* gene pairs were calculated using TBtools software. The *Ka*/*Ks* value can demonstrate whether the select pressure functions on protein coding *MtSDRs*. The *Ka*/*Ks* value of all the paralogous *MtSDR* gene pairs was below 1 (Appendix A), implying that purifying selection existed in the gene pairs.

### 2.5. Analysis of the Expression Pattern of MtSDRs in Different Organs

The spatial expression profile was analyzed based on microarray data from *M. truncatula* Gene Expression Atlas (MtGEA, https://mtgea.noble.org/v3/, accessed on 5 April 2021) of 156 *MtSDR*s in flowers, leaves, petioles, pods, roots, stems, and seeds (Appendix A), while the information of 57 *MtSDR* genes was not available. Above half of *MtSDRs* were expressed in at least one organ. Transcript of most of the genes in the *SDR108E* family was observed in roots (Figure 5). Some *MtSDRs* genes exhibited specific tissue expression. *MtSDR1E6, 65C7, 65C11, 114C3,* and *114C7* were preferentially expressed in flowers; *MtSDR460A14, 108E19, 114C15,* and *110C2* were specifically expressed in roots; while *MtSDR57C3* and *MtSDR119C3* were only expressed in seeds. The transcript levels of *MtSDR57C3* and *MtSDR119C3* were increased from 12 to 24 d, but decreased at 36 d after pollination, indicating that they might be associated with seed development.

### 2.6. Analysis of Cis-Acting Elements in the Promoter Region of MtSDRs

The cis-acting elements in the 2000bp sequence of the promoter region were analyzed using PlantCARE software. Thirteen types *cis*-acting elements associated with responses to stresses and phytohormones including wound responsive element (WUN-motif and WRE3), circadian responsive element (Circadia), drought responsive element (MBS), low-temperature responsive element (LTR), auxin responsive element (TGA-element and AuxRR-core), gibberellin responsive element (TATC-box, GARE-motif, and P-box), salicylic acid responsive element (TCA-element), MeJA responsive element (CGTCA-motif and TGACG-motif), and ABA response element (ABRE) were found in the promoters (Figure 6, Appendix A). A total of 280 wound responsive elements, 43 circadian responsive elements, 137 drought responsive elements, 89 low-temperature responsive elements, 128 auxin responsive elements, 151 gibberellin responsive elements, 131 salicylic acid responsive elements, 500 MeJA responsive elements, and 435 ABA responsive elements were identified in all *MtSDR* genes, indicating that *MtSDRs* participated in plant growth, development, and stress responses.

### 2.7. Expression Pattern of MtSDRs in Response to Abiotic Stresses

*MtSDRs* genes in five subfamilies including *SDR108E* (dihydroflavonol 4-reductase), *SDR110C* (ABA2 xanthoxin dehydrogenase family), *MtSDR7C* (chloroplast protein import translocon), *MtSDR65C* (tropinone reductase), and *MtSDR114C* (menthone/salutaridine reductase) in response to salt and drought were analyzed. The expression data were obtained from *M. truncatula* Gene Expression Atlas (MtGEA, https://mtgea.noble.org/v3/, accessed on 5 April 2021) (Appendix A). Of the 16 genes in the *MtSDR114C* family, 12 were upregulated after 6 h of salt treatment, and three were upregulated after 48 h (Figure 7). Of the 11 genes in *MtSDR110C* families, six were upregulated after 6 h of salt treatment, three were upregulated after 48 h, and two were downregulated. Most of the genes in *SDR108E* and *SDR65C* families were upregulated after 6 h or 48 h of salt stress, while transcript levels of nine genes (75%) in the *MtSDR7C* family were decreased after salt stress. In addition, most genes in the *MtSDR114C* family were upregulated by drought in roots, followed by downregulation after rewatering, while most genes in *MtSDR460A* family were downregulated after drought treatment, followed by upregulation after rewatering.

In order to understand the response of *MtSDR* genes to the cold, six *MtSDR* genes containing LTR *cis*-acting elements in the promoter region were analyzed. The results showed that *MtSDR65C8* and *Mt132C6* transcripts were increased after 6 h of cold treatment and reached a peak at 12 h (Figure 8). *MtSDR7C6* and *Mt110C15* transcripts were induced after 12 h of cold treatment, *MtSDR108E18* transcript was induced after 24 h, while *MtSDR108E3* transcript was greatly reduced after 2 h of cold treatment. The results implied that the *MtSDR* genes were cold-responsive.

### 2.8. Subcellular Localization of MtSDRs

The salt or drought responsive genes MtSDR7C16, 65C9, 108E24, 110C17, and 114C22 were selected for investigation for protein subcellular localization. GFP fluorescence signal was disseminated throughout the cell, while the signal of PM-marker protein was localized on the plasma membrane (Figure 9). The signal of MtSDR7C16, 65C9, 108E24, 110C17, and 114C22 fused with GFP was not co-localized with plasma membrane localization protein AtAKT1, indicating that MtSDR7C16, 65C9, 108E24, 110C17, and 114C22 were localized in the cytoplasm.

## 3. Discussion

The SDR superfamily is found in all organisms. The SDR superfamily has a conservative “Rossmann fold” structure and shows low sequence similarity among members, which makes it difficult to identify [26,27,28]. 213 *MtSDR* genes were identified in *M. truncatula* using the HMM models and similarity searches, which is similar to that in rice (227 *OsSDRs*) and maize (230 *ZmSDRs*) but different from that in *G. max* (315 *GmSDRs*, *A. thaliana* (178 *AtSDRs*)) [5]. The analysis revealed that MtSDR proteins had diverse variations in sequence length, molecular weight, and isoelectric points.

Ten conserved motifs were identified in MtSDR proteins. Among them, motif 1 and motif 2 that encode the conserved cofactor-binding site and the conserved active site existed in almost all MtSDR proteins. *MtSDR* genes were irregularly distributed on chromosomes. Homologous gene pairs were found in a few *MtSDRs*, indicating that the members of the *MtSDR* superfamily had low pairwise sequence identity. Compared to *Arabidopsis*, strikingly, based on the syntenic map of *M. truncatula* and *G. max*, 116 *MtSDR*s were shown to have 212 corresponding *G. max* orthologs. Therefore, the expansion of the *MtSDR* gene family may have appeared before the separation of *G. max.* The process of genetic evolution was probably reestablished by analyzing collinearity of the genomic sequences of SDR family in the same genome or between different genomes [29].

Gene functions are closely related to tissue-specific expression. Most of the genes in the *SDR108E* family were highly expressed in root, suggesting that *SDR108E* may be important for the function of roots. *MtSDR67E5* was highly expressed in all test tissues. *MtSDR67E5* is homologous to UDP-D-apiose/UDP-D-xylose synthases *(AXS2)* confers synthesis of Rhamnogalacturonan-II (RG-II), which is one of major pectin types in *M. truncatula* [30]. SDR proteins were involved in many processes of primary and secondary metabolism [31]. DVR (3,8-divinyl protochlorophyllide an 8-vinyl reductase) was an enzyme belonging to the extended SDR fold, participating in synthesis of monovinyl derivatives of chlorophyll in *Arabidopsis* [32]. Two NAD-dependent 3β-HSD/D (*At1g47290* and *At2g26260*), belonging to the *SDR31E* family, are involved in sterol biosynthesis in *Arabidopsis* [12,33]. The members in *SDR31E* family in *M. truncatula* are homologous to those in *SDR31E* family in *Arabidopsis*, implying that they might have similar functions.

The potential role of *MtSDR* genes in abiotic stress responses was analyzed in the study. Regulatory elements in the promoter region are associated in the regulation of gene expression, being temporal, spatial, and cell-specific [34]. The *cis*-elements responsive to abiotic stresses and hormones were identified in the promoter regions of *MtSDRs*. Plants can alleviate the damage caused by stress by enhancing the accumulation of secondary metabolites [31]. Remarkably, the four families responsive to abiotic stress were involved in secondary metabolism: *MtSDR65C* (tropinone reductase), *SDR108E* (Dihydroflavonol 4-reductase) family, *SDR110C* (ABA2 xanthoxin dehydrogenase family), and *MtSDR114C* (menthone/salutaridine reductase). The conversion of xanthoxin to abscisic aldehyde was catalyzed by ABA2 (SDR1) in *Arabidopsis*. *MtSDR110C2*, the homolog of *AtSDR1*, was highly expressed in roots, suggesting that it may also involve in the conversion of ABA [15]. Dihydroflavonol 4-reductase (*SDR108E*) was one of the key regulatory enzymes of flavan-3-ols biosynthesis and protects plants against oxidative damage in transgenic tobacco [35]. Members in *MtSDR7C, MtSDR65C, MtSDR110C, MtSDR114C,* and *MtSDR108E* families were up- or down-regulated in response to salt and drought stresses, while the expression of *MtSDR7C6*, *Mt65C8*, *Mt108E18*, *Mt110C15,* and *Mt132C6* genes was altered after cold treatment. The results suggest that MtSDRs may be associated with abiotic stress adaptation. The above genes should be selected for further studies on the role of abiotic stress tolerance in *M. truncatula.*

## 4. Materials and Methods

### 4.1. Identification of MtSDR Genes

Arabidopsis SDR protein sequences (178) were used as baits to search for the target sequence of *M. truncatula.* The sequences of *SDR* genes were gained from genome databases of *Arabidopsis* (TAIR, http://www.arabidopsis.org/, accessed on 1 April 2021) and *M. truncatula* (http://www.medicagohapmap.org/, accessed on 1 April 2021), respectively. The candidate SDR-encoding genes were chosen and annotated based on the closest *A. thaliana* homology. Three Pfam HMMs were used to select the genome set of predicted proteins: PF00106, PF01370, and PF01073 using HMMER3 (http://hmmer.org/, accessed on 5 April 2021) [25]. The decision rules for SDR inventory were employed as described by Moummou et al. [5]. The MtSDRs protein sequences were predicted using SMART (http://smart.embl-heidelberg.de/, accessed on 5 April 2021) [36] and InterPro (http://www.ebi.ac. uk/interpro/, accessed on 5 April 2021), while the redundant sequences were removed. The physicochemical properties of MtSDRs including protein sequence length (SL), molecular weight (MW), and isoelectric point (PI) were analyzed using the ExPasy online tool (http://web.expasy.org/protparam/, accessed on 5 April 2021).

### 4.2. Phylogenetic Analysis and Multiple Sequence Alignment

The multiple sequence alignment of the MtSDR domains was analyzed using ClustalW algorithm [37]. Phylogenetic analysis of SDRs in *M. truncatula* was conducted using MEGA-X with the maximum-likelihood (ML) method in 1000 bootstrap replicates [38].

### 4.3. Motif Analysis

MEME (http://meme.nbcr.net/meme/intro.html, accessed on 5 April 2021) was applied to analyze SDR conserved motifs [39]. The results were presented using TBtools software [40].

### 4.4. Chromosomal Distribution and Synteny Correlation Analysis

The circos diagram was illustrated using the genome annotation file for analysis of chromosomal locations of *MtSDR* genes. The physical position of the *MtSDR* genes on the chromosome was mapped using TBtools software. The synteny correlation analysis of *SDR* genes between the homologs in *M. truncatula* and *A. thaliana* or *G. max* were verified and visualized using TBtools software.

### 4.5. Analysis of Microarray Expression Profile

Microarray data on the expression profile of *MtSDR* genes in different tissues and responses to drought and salt were extracted from the MtGEA (https://mtgea.noble.org/v3/, accessed on 5 April 2021). The heatmap was drawn using the TBtools software.

### 4.6. Analysis of cis-Acting Regulatory Elements

The promoter sequences (2000 bp) upstream of the transcription start site of all MtSDRs were downloaded from the *M. truncatula* genome database and analyzed using PlantCARE software (http://bioinformatics.psb.ugent.be/webtools/plantcare/html/, accessed on 6 April 2021) [41].

### 4.7. Analysis of MtSDRs Expression in Response to Cold

*M. truncatula* (Jemalong) A17 seedlings were grown in a greenhouse (16 h light/8 h dark, 20 to 28 °C) for four weeks as previously described [42]. The plants were transferred to a growth chamber at 4 °C for 24 h under 12 h light for cold treatment. Leaves (0.1 g) were frozen in liquid nitrogen and used for extraction of total RNA and synthesis of cDNA using the TIANGEN total RNA Kit and TaKaRa PrimeScript RT kit with gDNA Eraser, respectively, as described by the manufacturer’s instructions. Primers were designed using the PrimerQuest tool (http://www.idtdna.com/Primerquest/Home/Index, accessed on 6 April 2021) and listed in Appendix A. qRT-PCR was conducted using Thermal Cycler Dice Real Time System II (Takara, Japan). The *MtActin*7 (*Medtr3g095530*) gene was amplified as the internal control to normalize the amount of the template. Relative expression was calculated using the normalized (2^−∆∆*Ct*^) value.

### 4.8. Analysis of Subcellular Localization

The coding sequences of several *MtSDR**s* without terminal codon were amplified and fused with GFP by cloning to vector pCAMBIA-1305, being driven by the CaMV 35S promoter. The constructed vector or empty vector as control was co-transformed with 35S::mCherry vector or 35S::*AtAKT1*-mCherry vector to the abaxial surface of 4-week-old *N. benthamiana* leaves using an *Agrobacterium*-mediated method. Fluorescence was observed 72 h after transformation using a confocal laser scanning microscope (Zeiss LSM800, Germany).

## 5. Conclusions

A total of 213 MtSDR proteins were identified from *M. truncatula*. They were classified into three primary groups (classical SDR fold, atypical members, and extended SDR fold). Segmental duplication was the crucial driving strength for *MtSDRs* evolution. In addition, MtSDR proteins have undergone strong purification pressures during their evolution. Above half of *MtSDR*s were expressed in at least one organ, and lots of *MtSDRs* had a preference for tissue expression. The *cis*-acting element responsive to plant hormones (salicylic acid, ABA, auxin, MeJA, and gibberellin) and stresses were found in the promoter of some *MtSDRs*. Many genes of *MtSDR*7*C,*
*MtSDR*65*C**, MtSDR*110*C**, MtSDR*114*C*, and *MtSDR108E* families were responsive to drought, salt, and cold. The study provides useful information for further investigation on biological functions of MtSDRs, especially in abiotic stress adaptation, in the future.

## Figures and Tables

**Figure 1 ijms-22-09498-f001:**
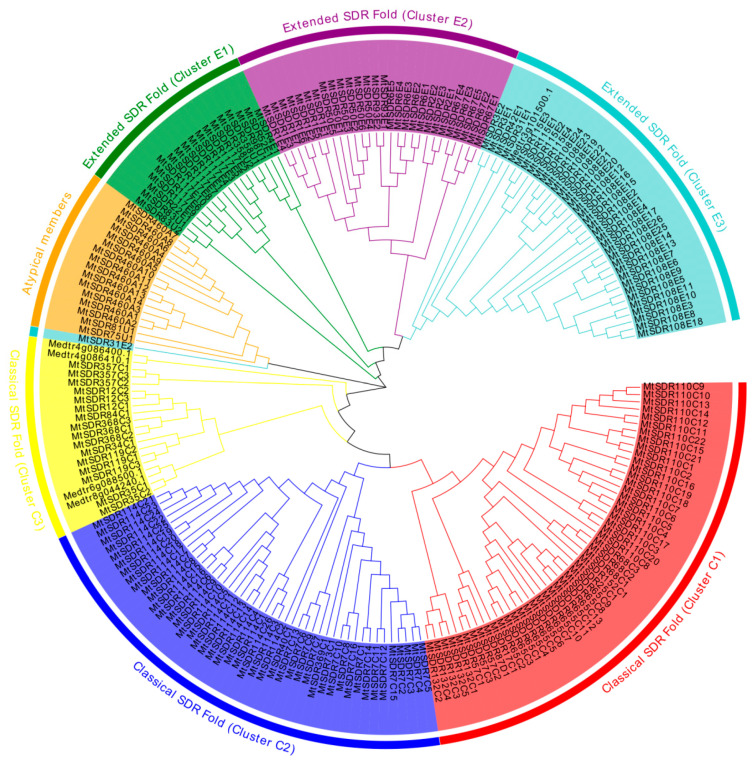
Phylogenetic tree of the domain relationship of MtSDRs. The different colors perform the different groups (or subgroups) of MtSDR.

**Figure 2 ijms-22-09498-f002:**
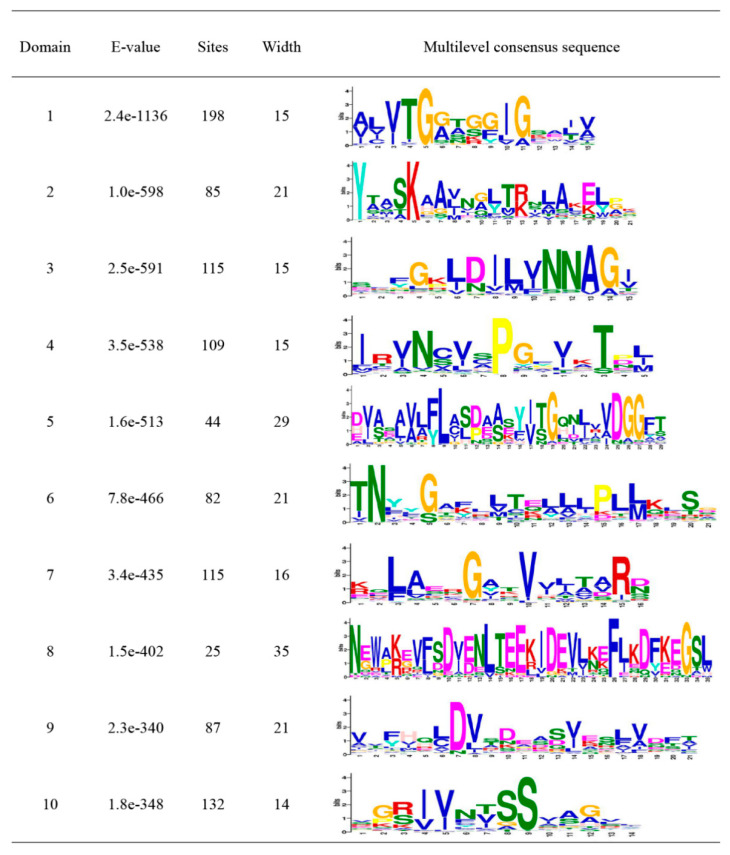
Details of Ten Motifs of MtSDRs.

**Figure 3 ijms-22-09498-f003:**
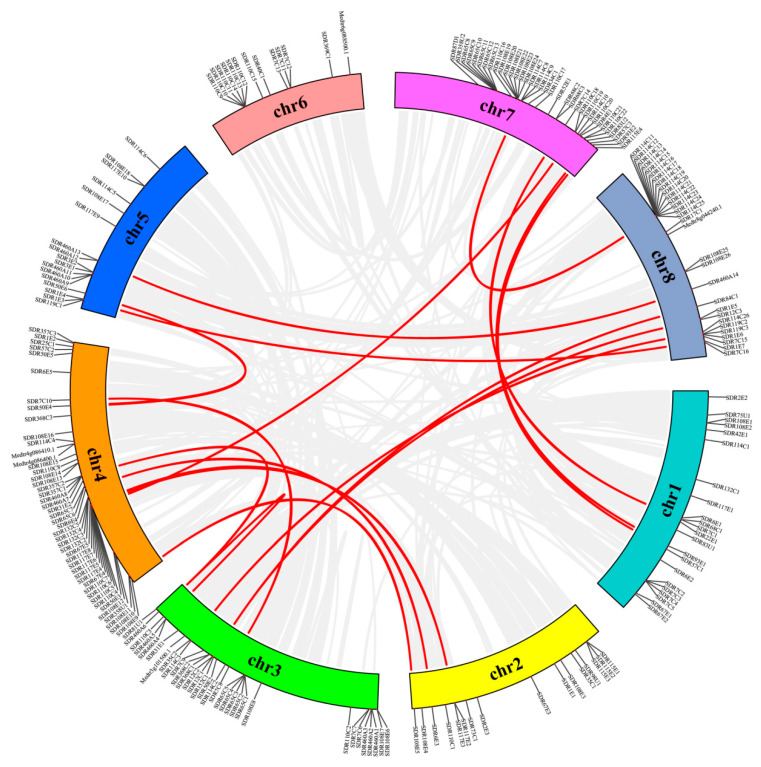
Synteny analysis of interchromosomal relationships of *MtSDR* genes. Red lines indicate duplicated gene pairs.

**Figure 4 ijms-22-09498-f004:**
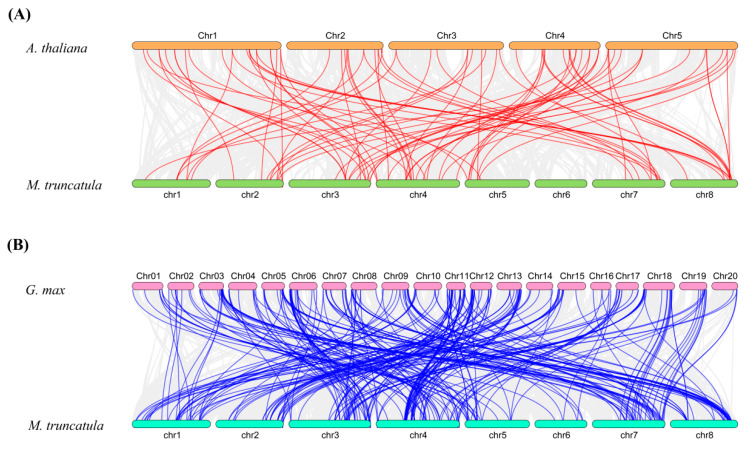
Synteny analysis of *MtSDR* genes in the genomes between *M. truncatula* and *A. thaliana* or *G.max*. (**A**) *M. truncatula* and *A. thaliana*; (**B**) *M. truncatula* and *G.max.* The gray lines show collinear blocks. The red and blue lines indicate the syntenic gene pairs between *M. truncatula* and *A. thaliana* or *G. max*, respectively.

**Figure 5 ijms-22-09498-f005:**
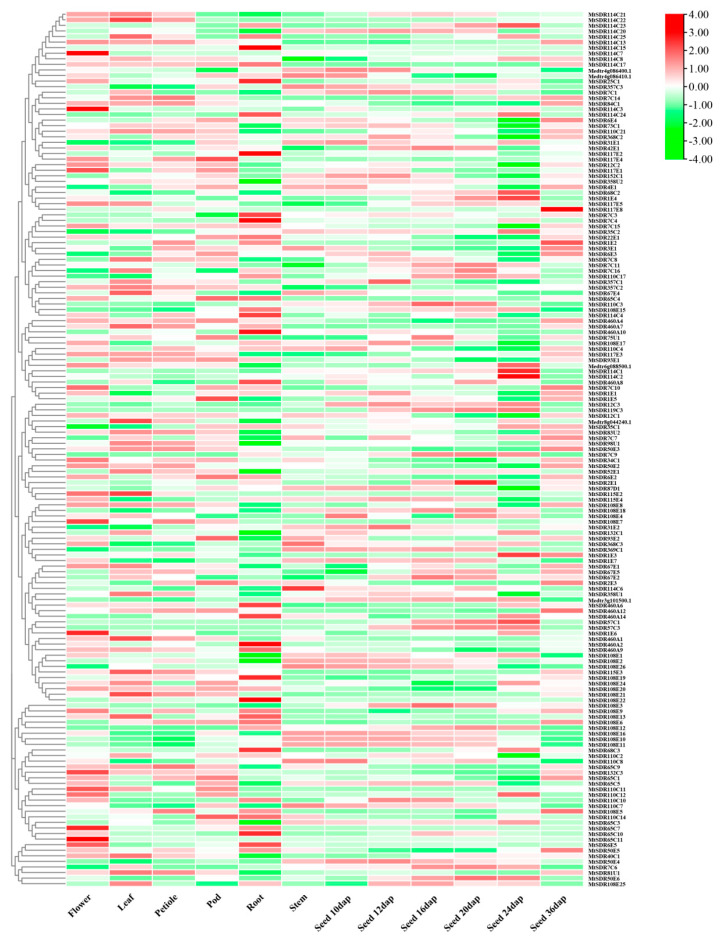
Heat map of *MtSDRs* in different organs. The expression data of *MtSDR**s* in flowers, leaves, petioles, flowers, roots, stems, and seeds were collected from *M. truncatula* Gene Expression Atlas (MtGEA, https://mtgea.noble.org/v3/, accessed on 5 April 2021). The relative expression of heat map was characterized by log_2_ transformed.

**Figure 6 ijms-22-09498-f006:**
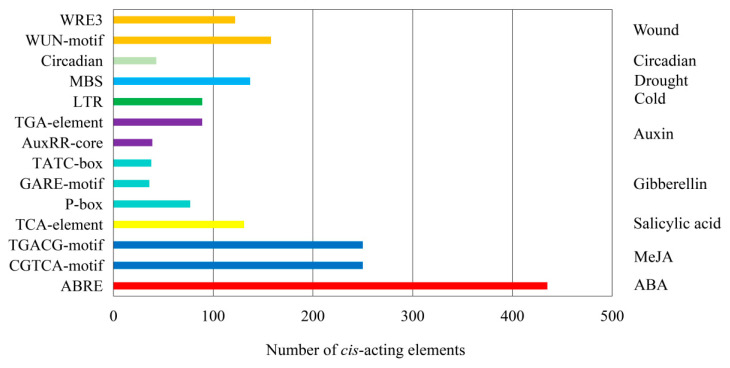
The number of *cis*-acting elements was tested in the promoter region of *MtSDRs*. The name of *cis*-acting elements was listed on the left side of the image, and the corresponding function annotation was listed on the right side.

**Figure 7 ijms-22-09498-f007:**
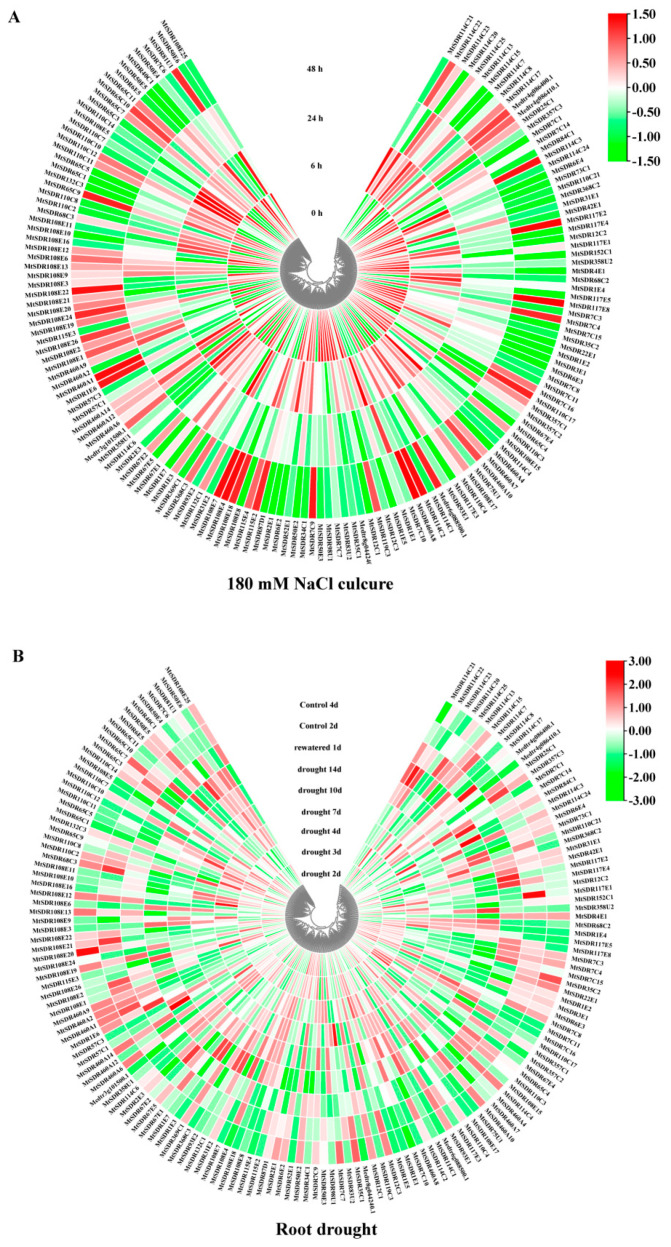
Heat map of *MtSDR*s expression in response to salt and drought stress. The expression data were collected from *M. truncatula* Gene Expression Atlas (MtGEA, https://mtgea.noble.org/v3/, accessed on 5 April 2021). (**A**) Two-week-old seedlings were treated with 180 mM NaCl for 0, 6, 24, and 48 h, respectively. (**B**) Twenty-four-day-old seedlings were withholding irrigation for soil drying for 14 d, followed by 1 d of rewatering.

**Figure 8 ijms-22-09498-f008:**
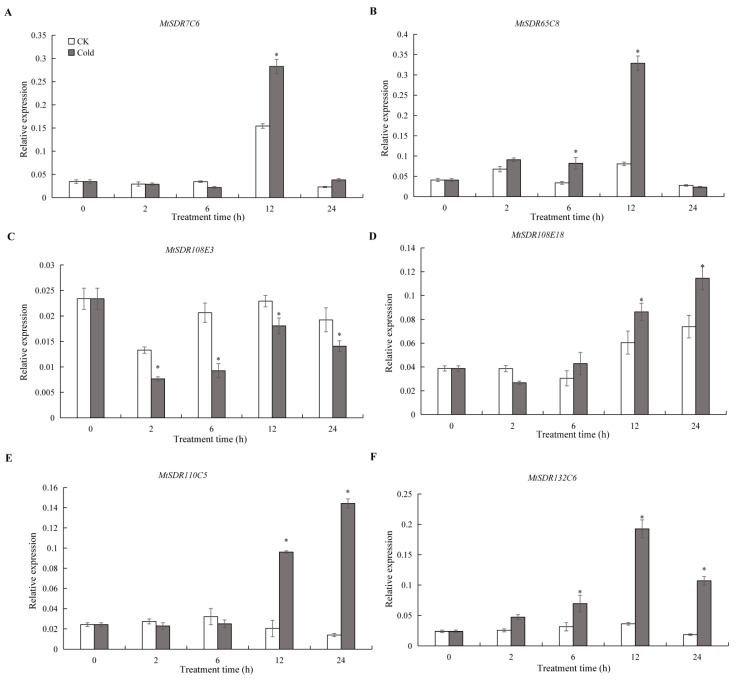
Relative expression of *MtSDR7C6* (**A**), *SDR65C8* (**B**), *SDR108E3* (**C**), *SDR108E18* (**D**), *SDR110C15* (**E**)*,* and *SDR132C6* (**F**) in response to cold treatment. Four-week-old seedlings were placed in a growth chamber at 4 °C for 0, 2, 6, 12, and 24 h. Mean values and standard errors were calculated from three biological replicates. * indicates significant difference between cold treatment and control using one-way ANOVA at *p* ≤ 0.05.

**Figure 9 ijms-22-09498-f009:**
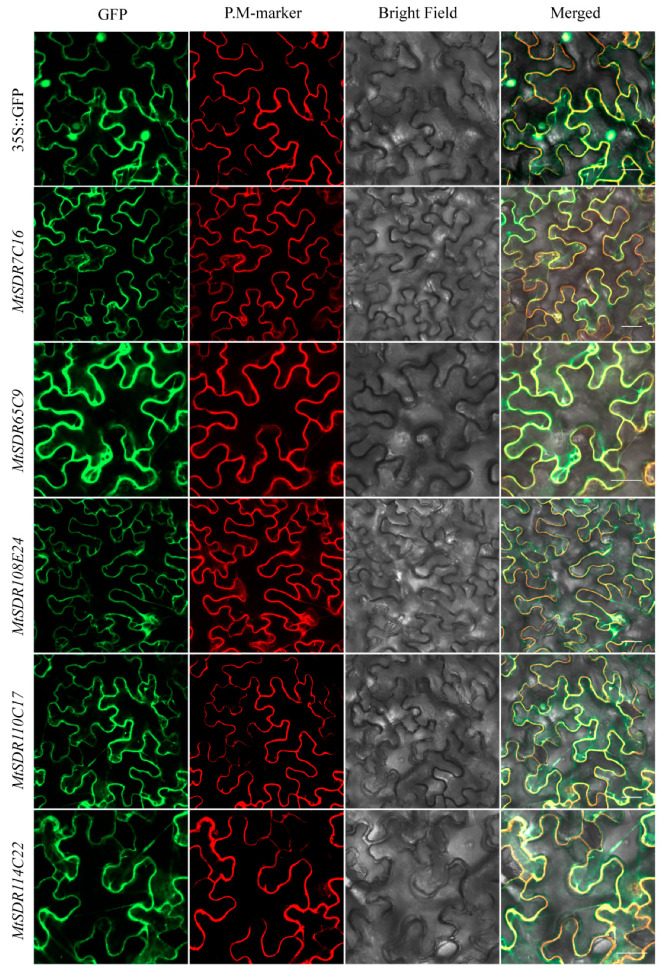
Subcellular localization of MtSDRs. The injected tobacco was used for observation. PM-marker: plasma membrane localization protein AtAKT1. Bars = 20 µm.

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
