# Peer review of "Genome-Wide Identification and Characterization of Short-Chain Dehydrogenase/Reductase (SDR) Gene Family in Medicago truncatula"

_ijms, 2021, doi:10.3390/ijms22179498_

Round 1

Reviewer 1 Report

All SDR genes in M. truncatula were identified and characterized comprehensively in this study. The basic characteristics of the MtSDRs were studied, including their phylogenetic relationship, gene structure, and common protein features. These findings suggested that genes from the MtSDR7C, MtSDR65C,  MtSDR110C, MtSDR114C, and MtSDR108E families might regulate abiotic stresses. The manuscript is well structured and well discussed. However, some points should be checked and corrected before its acceptance in this journal.  Therefore, I recommended the publications of the paper after major revision according to given my comments.

  1. Please try Cellular localization SDR genes in M. truncatula. If you add some Microscopy  results it will be good.
  2. Please speculate about the reasons for the obtained results. The discussion needs to improve.
  3. Please add conclusion section. In Conclusion, the authors should add the significance of this research, and its potential practical application.
  4. English of the MS needs to be greatly improved. The English of the whole article has to be checked carefully to eliminate linguistic errors.

Author Response

Response to the comments

Dear Editor,

Thank you for your decision and processing our manuscript (ijms-1344601).  The manuscript has been revised addressing all the comments.  In addition to the scientific points, language has been polished throughout the text. However, we did not use the “Track Changes” function to show the revisions because there are too many errors in spelling or in each paragraph. Followed are the responses to the comments point to point.

Sincerely yours,

Zhenfei Guo

Reviewer 1

Point 1: Please try Cellular localization SDR genes in M. truncatula. If you add some Microscopy results it will be good.

Response: The subcellular localization of some SDR proteins including MtSDR7C16, Mt65C9, Mt108E24, Mt110C17 and Mt114C22 has been completed and added in the revision (Fig. 9).

Point 2: Please speculate about the reasons for the obtained results. The discussion needs to improve.

Response 2: Discussion has been revised.

Point 3: Please add conclusion section. In Conclusion, the authors should add the significance of this research, and its potential practical application.

Response 3: The conclusion section has been added in the revision.

Point 4: English of the MS needs to be greatly improved. The English of the whole article has to be checked carefully to eliminate linguistic errors.

Response 4: The manuscript has been checked carefully and almost rewritten.

Finally, we appreciate very much for your suggestions and comments, which are valuable in improving the quality of our manuscript.

Thank you and best regards.

Reviewer 2 Report

Dear Authors,

The manuscript reports a highly interesting, original research work. The conclusions are clear, however I suggest a minor change in the description of methods.

- Please carefully separate in the 'Methods' section which data were available from a public database and which were generated from your own work, especially in the case of expression patterns, e.g.: expression patterns regarding salt and drought stress are based on available data while expression pattern under cold stress originated from this work. - Please strictly write the details of your cold stress experiment (How many plants were used, which organs were analysed, etc.?)   And a general terminological note: the genes can not regulate 'stress', only the 'stress response' affected by 'stress factor' (e.g. in lines 22 and 292-294). Please revise these sections.

Author Response

Response to the comments

Dear Editor,

Thank you for your decision and processing our manuscript (ijms-1344601).  The manuscript has been revised addressing all the comments.  In addition to the scientific points, language has been polished throughout the text. However, we did not use the “Track Changes” function to show the revisions because there are too many errors in spelling or in each paragraph. Followed are the responses to the comments point to point.

Sincerely yours,

Zhenfei Guo

Reviewer 2

Point 1: Please carefully separate in the 'Methods' section which data were available from a public database and which were generated from your own work, especially in the case of expression patterns, e.g.: expression patterns regarding salt and drought stress are based on available data while expression pattern under cold stress originated from this work.

Response 1: The details which data were collected from database and which were performed by the authors have been presented in Materials and Methods section and in legends.

Point 2: Please strictly write the details of your cold stress experiment (How many plants were used, which organs were analysed, etc.?).

Response 2: Cold treatment and the tissue used for isolation of total RNA have been stated in the Materials and Methods section.

Point 3: And a general terminological note: the genes can not regulate 'stress', only the 'stress response' affected by 'stress factor' (e.g. in lines 22 and 292-294). Please revise these sections.

Response 3: The manuscript has been rewritten, the errors have been revised.

Finally, we appreciate very much for your suggestions and comments, which are valuable in improving the quality of our manuscript.

Thank you and best regards.

Round 2

Reviewer 1 Report

Requested corrections have been completed.

This manuscript is a resubmission of an earlier submission. The following is a list of the peer review reports and author responses from that submission.